# Differences in Aroma Profile of Cabernet Sauvignon Grapes and Wines from Four Plots in Jieshi Mountain Region of Eastern China

**DOI:** 10.3390/foods12142668

**Published:** 2023-07-11

**Authors:** Zhuo Chen, Yang-Peng Wu, Yi-Bin Lan, Yan-Zhi Cui, Tong-Hua Shi, Yu-Bo Hua, Chang-Qing Duan, Qiu-Hong Pan

**Affiliations:** 1College of Food Science and Nutritional Engineering, China Agricultural University, Beijing 100083, China; 2Key Laboratory of Viticulture and Enology, Ministry of Agriculture and Rural Affairs, Beijing 100083, China; 3Bodega Longes Co., Ltd., Qinghuangdao 066600, China

**Keywords:** Cabernet Sauvignon, aroma, wines, plots, differences

## Abstract

The Bohai Bay region is a famous wine-growing area in China, where the rainfall is concentrated in the summer due to the influence of the temperate semi-humid monsoon climate. As such, the vineyard terrain has a significant impact on the flavor quality of the grapes and the resulting wines. To explore the relationship between the ‘Cabernet Sauvignon’ wine style and terrain, this study takes four different plots in the Jieshi Mountain region to investigate the differences in the aroma profile of Cabernet Sauvignon grapes and wines of two consecutive vintages. Based on two-way ANOVA, there were 25 free and 8 glycosylated aroma compounds in the grapes and 21 and 10 aroma compounds with an odor activity value greater than 0.1 in the wines at the end of alcohol fermentation (AF) and malolactic fermentation (MLF), respectively, that varied among the four plots. Wines from the four plots showed a significant difference in floral and fruity aroma attributes, which were mainly related to esters with high odor activity values. The difference in concentration of these compounds between plots was more pronounced in 2021 than in 2020, and a similar result was shown on the Shannon–Wiener index, which represents wine aroma diversity. It has been suggested that high rainfall makes the plot effect more pronounced. Pearson’s correlation analysis indicated that concentrations of *(E)*-3-hexen-1-ol in grapes and ethyl 3-methylbutanoate, ethyl hexanoate, isoamyl acetate, isopentanoic acid, and phenethyl acetate in wines were strongly positively correlated with the concentrations of N, P, K, Fe, and electrical conductivity in soil but negatively correlated with soil pH. This study laid a theoretical foundation for further improving the level of vineyard management and grape and wine quality in the Jieshi Mountain region.

## 1. Introduction

Wine quality is closely linked to the quality of the grape, which largely determines the color, aroma, and flavor of the wine and influences the formation of the fermented aroma profile. Grape quality is influenced by many factors. Studies have shown that climate has the greatest influence on the composition and quality of grapes and wines, followed by soil characteristics, which are able to buffer unfavorable vintage effects even within a small wine region [1,2,3]. In the same region, under the same climate conditions, the variation in the quality of wine produced from different plots is largely due to the variation in soil, including soil texture [4], soil nutrients [5], etc.

Different styles of wine can be produced from different parts of the same vineyard when under uniform management [6]. Bramley et al. studied the terroir conditions and wine quality of different plots in a vineyard in the Murray Valley region and demonstrated a strong correlation between the two [7]. The study of terroir between different plots of the same vineyard can help to improve understanding of the factors that affect grape and wine quality, which is of great importance for accurate vineyard management and improvement of grape and wine quality.

The quality of wine is mainly judged by indicators such as color, aroma, mouthfeel, and aftertaste [8]. Among these indicators, aroma is very important in evaluating the flavor quality of wine and determining the differences between wine styles around the world, thereby influencing consumer preferences [9]. Many factors influence wine aroma, including the growing environment (climate, soil, and light), raw and auxiliary materials (grape varieties and yeast strains), and the winemaking process (fermentation and ageing) [10]. Reynolds et al. investigated the correlation between the spatial distribution of terroir and grape aroma in a Canadian ‘Riesling’ vineyard in Ontario and found that soil texture and nutrients were related to the berry weight and grape terpenes [11]. ‘Pinot noir’ grapes grown in different regions but under standardized winemaking conditions produced wines with unique chemical and sensory profiles, which generally persisted through ageing, and soil pH may be one of the important factors [12].

Research on terroir in China is mainly conducted in the wine regions of the eastern foothills of the Helan Mountains in Ningxia and the northern foothills of the Tianshan Mountains in Xinjiang, focusing on the influence of soil conditions on grape fruit quality [13,14]. Peng et al. analyzed grape aroma compounds from different plots in the eastern foothills of the Helan Mountains in China and found that the contents of C6/C9 compounds, esters, C13-norisoprenoid and terpene were the significant compounds between different plots [14]. Zhang et al. investigated the influence of environmental factors on the physical and chemical parameters of wine produced in the Jieshi Mountain region [15]. Ling et al. identified the styles of white wines produced in the Jieshi Mountain region [16]. The Jieshi Mountain region is located in Changli County, northeastern Hebei Province (39°43′–39°83′ north latitude), bordered by Bohai Bay to the east, Yanshan Mountains to the north, and the Luan River to the southwest, forming a unique regional climate characterized by mountains, seas, and rivers. The grape-growing region has a temperate semi-humid monsoon climate, with an average annual rainfall of 600–650 mm mainly concentrated in July, August, and September, which makes the terrain of the vineyard have a significant impact on the grape berry quality. As an old producing area in China, research on plot influence on grape and wine flavor in Jieshi Mountain region remains limited.

In this study, four plots of a winery in the Jieshi Mountain region, Qinhuangdao, China, were all located on the same slope. Due to the different aroma qualities of the wines made from the grapes of the four plots, it is speculated that the geographical location and soil characteristics may be related, but the specific relationship between the two remains unclear. Based on aroma compound data from 2020 to 2021, this paper investigates the relationship between aroma differences of grapes and wine among four plots and terroir conditions (soil conditions and meteorological conditions), and studies the influence of terroir conditions on the formation of aroma profiles of grapes and wine. The aim of this study is to provide the wine region with a theoretical basis for carrying out fine management of vineyards according to the characteristics of the plots in order to improve wine quality.

## 2. Materials and Methods

### 2.1. Materials

#### 2.1.1. Experimental Site

Four plots of *Vitis vinifera* L. cv. Cabernet Sauvignon at Bodega Langes in the Jieshi Mountain region were evaluated over two vintages (2020 and 2021), namely CS1, CS2, CS3, and CS4, respectively (Appendix A). Plot CS1 was located in the northwestern area with an area of 8.54 ha planted in 2001. Plot CS2 was located in the northeast and east area with an area of 5.69 ha planted in 2004. Plot CS3 was located in the central area with an area of 4.94 ha planted in 2006. Plot CS4 was located in the eastern area with an area of 7.14 ha planted in 2011. All plots were located on the same slope with a gradient of 0.3%, with CS1 at the top of the slope and CS2 at the bottom. All the vines were trained to a sloping trunk with a vertical shoot-positioning trellis system, with spacing of 2 m × 1 m and rows planted in a north–south orientation.

#### 2.1.2. Soil Sampling and Analysis

Soil samples were collected using a 9-point sampling method, with these points distributed in a Z-shape in each plot, each point being collected at a distance of 50–70 cm away from the vine in the row, at three depths: 0–30 cm, 30–60 cm, and 60–90 cm. Soils collected at the same depth from the nine sites were completely mixed and then divided into three replicates for analysis of particle content, organic matter, electrical conductivity, pH, and cation exchange capacity (CEC). Soil from the 30–60 cm depth, which is the root enrichment zone, was used for analysis of soil mineral elements. The determination of basic soil physico-chemical properties was based on Han [17]: Soil pH was measured in KCl solution with a soil/solution ratio of 1:2.5 *v*:*v*; organic matter was determined via sulfochromic oxidation; electrical conductivity (EC) was measured with a conductivity meter; and CEC was determined via the ammonium acetate method [18]. N was determined via the Kjeldahl method, which consists of three steps: sample digestion, distillation, and ammonia determination [19]. P, Fe, Ca, K, and Mg contents were determined via inductively coupled plasma mass spectrometry (ICP-MS) (Agilent 7800, Santa Clara, CA, USA). First, 0.25 g of soil was taken, and 5 mL of HNO_3_ was added for digestion. Secondly, the solution was heated at 100 °C for 30 min before cooling. After cooling and heating until nearly dry, 1 mL of H_2_O_2_ was added. After cooling, the double-distilled water volume was reduced to 50 mL and analyzed. The ICP-MS was equipped with an autosampler, a Burgener nebulizer, nickel cones, and a peristaltic sample delivery pump. Detection parameters were as follows: 15 L/min plasma gas flow, 4.3 mL/min helium and reaction gas flow, 0.90 L/min carrier gas flow (>99.99% argon purity), 0.3 r/s sample lift rate, and an atomization chamber temperature at 2 °C. An external standard method was used for quantification, which was prepared with a multi-element standard solution (ICP-MS-CAL2-1, AccuStandard, New Haven, CT, USA) in 0.5% HNO_3_ (chromatographically pure) as described by Wu [20].

#### 2.1.3. Grape Berry Sampling

The 5-point sampling method was used for berry sampling. The Cabernet Sauvignon grapes were commercially harvested on 6 October 2020 and 2 October 2021, respectively. During sampling, six berries were randomly selected from the upper, lower, left, right, front, and back positions of different clusters, and the shade and sunny sides of each row were uniformly and randomly sampled. A total of 600 grapes were collected, of which 100 were used for the analysis of physico-chemical analysis, and the rest were frozen in liquid nitrogen and stored at −80 °C for the analysis of aroma compounds.

### 2.2. Reagents and Equipment

Chromatographically pure grade dichloromethane, methanol, and ethanol were purchased from Honeywell, USA. Analytical grade glucose, sodium hydroxide, sodium chloride, citric acid, malic acid, tartaric acid, and sodium dihydrogen phosphate were purchased from Beijing Chemical Reagent Company. Volatile standards and N-Alkanes (C6-C24) were purchased from Sigma-Aldrich. Yeast Zymaflore FX10, pectinase LAFASE HE GRAND CRU, and Lactobacillus B7 DIRECT 25HL were purchased from LAFFOTA, France.

### 2.3. Methods

#### 2.3.1. Acquisition of Meteorological Data

Meteorological data were obtained from the self-built weather station in the vineyard of Bodega Langes. Temperature, humidity, rainfall, wind direction, and weather were recorded every 8 h (8:00, 16:00, 24:00). Meteorological data at the vineyard were recorded from 2020 to 2021 (Appendix A). It was found that the effective accumulated temperature (from April to September and from August to September) for grapevines was significantly higher in 2020 than in 2021. By contrast, rainfall (from June to August) in 2021 was significantly higher than in 2020. The rainfall in September was only 71.6 mm in 2020, while it was 193.1 mm in 2021. The number of rainy days in September 2021 was 8 more than in the same month in 2021. In terms of monthly indicators (Appendix A), the average monthly temperature (from June to August) in 2020 was significantly higher than in 2021, and the average monthly humidity in most months in 2021 was significantly higher than in 2020. Rainfall and number of rainy days in September (harvest period) in 2021 were 2.7 and 3.7 times higher than in 2020, respectively. The number of sunny days per month from April to September in 2020 was larger than in 2021. In general, the weather conditions in 2020 were more suitable for the growth of grapes than those in 2021.

#### 2.3.2. Small-Scale Winemaking Procedure

Grapes from four plots were fermented separately, and two 300-liter stainless steel fermenters were prepared for each plot. A standard winemaking procedure was followed for all wines. For each fermenter, 240 kg of grapes were destemmed and crushed, with the addition of 6% sulfite to give a final concentration of 55 mg/L sulfur dioxide, and then stirred evenly. Zymaflore FX10 yeast (200 mg/L) and LAFASE LE GRAND CRU pectinase (40 mg/L) were added. Fermentation was carried out at 22–25 °C. Three 200-mL bottles of grape juice were collected before the addition of yeast and pectinase and stored at −20 °C for later analysis. During fermentation, the must was stirred with a cap press every 8 h, and the specific gravity and temperature of the must were monitored. At the end of alcoholic fermentation, the residue was separated from the wine, and three bottles of wine (750 mL each) were collected from each fermenter. The separated free-run wine was transferred in its entirety to a 50-liter vessel and inoculated with B7 DIRECT 25HL lactic acid bacteria. After malolactic fermentation, six bottles of wine sample (750 mL each bottle) were collected for each vessel. In 2020, alcohol fermentation and malolactic fermentation took 11 days and 13 days, respectively. In 2021, alcohol fermentation and malolactic fermentation took 10 days and 12 days, respectively.

#### 2.3.3. Determination of Basic Physico-Chemical Properties of Grapes and Wine

One hundred berries were randomly selected and weighed for hundred-grain weight, then the berries were squeezed for their juice for detection. The soluble solids of the juice were measured with a saccharometer and the pH with a pH meter. Titratable acid was titrated with NaOH and measured as tartaric acid (g/L). Titratable acid was determined according to GB/T 15038-2006 ‘Analytical methods of wine and fruit wine’. The physico-chemical properties of the wine, including alcohol, reducing sugars, titratable acid, pH, and volatile acid, were detected using an OeneFoss wine analyzer (Foss Ltd., Hilleroed, Denmark).

#### 2.3.4. Extraction and Detection of Grape Aroma Compounds

The extraction of free and bound aroma compounds was conducted according to the method of He et al. [21], 60–70 g of de-seeded grape berries were ground with 0.5 g D-gluconolactone and 1 g polyethylpyrrolidone (PVPP) in liquid nitrogen to prevent oxidation of the sample, then were macerated for 4 h at 4 °C and centrifuged at 8000 rpm for 10 min at 4 °C to obtain clear must. Bound aroma compounds were isolated using Cleanert PEP-SPE resins, and enzymatic hydrolysis of glycosidic precursors was conducted at 40 °C for 16 h with the addition of 100 μL AR 2000 (Rapidase, 100 g/L).

Headspace solid-phase microextraction-gas chromatography-tandem mass spectrometry (HS-SPME-GC-MS) was used to analyze the aroma compounds of grapes as described by Wen et al. [22]. Samples were prepared, each consisting of 5 mL of grape juice with an addition of 1 g of sodium chloride and 10 μL of 4-methyl-2-pentanol solution (internal standard). Samples were placed in a CTC-Combi PAL autosampler (CTC Analytics, Zwingen, Switzerland) equipped with a 2 cm DVB/CAR/PDMS 50/30 μm SPME fiber (Supelco, Bellefonete, PA, USA) and agitated at 500 rpm for 30 min at 40 °C. The SPME fiber was then inserted into the headspace to absorb aroma compounds at 40 °C for 30 min and was instantly desorbed into the GC injector to desorb the aroma compounds. Aroma compounds were analyzed with an Agilent gas chromatography-mass spectrometer (Agilent 6890 GC-5975C MS, Santa Clara, CA, USA). fitted with an Agilent 19091N-136hp-InnoWaxPolythyleneol capillary column (60.0 m × 0.25 mm × 0.25 μm). A 1 μL splitless automatic injection procedure was used, with helium as the carrier gas, at 1 mL/min, and the inlet temperature was 250 °C. Oven temperature began with 50 °C for 1 min and then increased to 220 °C at a rate of 3 °C/min and held for 5 min. The ion source (EI) temperature and mass spectrum interface temperature were set at 230 °C and 280 °C, respectively. The ionization energy was set at 70 eV, and the mass scan range was 30–350 u.

The qualitative and quantitative method of measuring aroma compounds followed the method described in our previous report [23]. The aroma compounds were qualified via comparison of the retention indices, mass spectrometry, and the NIST11MS database of analytes and standards. The concentrations of volatile compounds were expressed as μg/L in wines and μg/kg of fresh berry weight of grapes.

#### 2.3.5. Analysis of Aroma Compounds in Wine

HS-SPME-GC-MS was used to analyze aroma compounds in wine according to the method of Lan et al. [24]. The pretreatment and detection methods were almost the same as for grapes, except for the injection mode. The sample was injected in split mode, which was different from that used for grapes.

The qualitative and quantitative characteristics of wine aroma were the same as mentioned above.

#### 2.3.6. Sensory Evaluation of Wine

The quantitative description analysis method (QDA) was adopted for sensory evaluation as described by Lan et al. [25]. The evaluation team (19 persons, including 7 males and 12 females, aged from 22 to 30 years old) was composed of long-term trained evaluators. First, the descriptors were determined. The members of the evaluation team checked the descriptors of wine samples from the vocabulary of red wine descriptors and finally determined the unified descriptors with high check frequency after statistics, discussion, and analysis. The above descriptors were applied to the intensity evaluation of wine samples by using the 10-point system. The analysis results were displayed by the radar image after statistical analysis.

#### 2.3.7. Data Processing

The odor active value (OAV) of aroma compounds was calculated by dividing the concentration of the aroma compound in the sample (μg/L) by the olfactory threshold of the analyte detected in the water medium or simulated wine solution (μg/L). Microsoft Excel 2019 was used for data pre-processing. SPSS Statistic 20.0 was used for statistical analysis, Duncan’s method for one-way analysis of variance (ANOVA) with significance level *p* < 0.05, and Pearson’s method for linear relationship analysis. GraphPad was used for two-way ANOVA. Alpha diversity is often used for the analysis of biodiversity in systems biology [26]. In this study, the Shannon–Wiener index was used to evaluate the diversity of aroma compounds and the formula was modified as follows [27].
(1)$$\mathrm{Diversity}=-\sum_{i=1}^{N}P_{i}\ln P_{i}$$

In Formula (1), where Pi represents the relative concentration of VOCi and *N* represents the total number of VOC in a particular grape sample, Pi was calculated as follows: Pi = ni/*N*, where ni represents the concentration of VOCi and *N* represent the total concentration of VOCs in a particular grape sample, respectively.

Triangular diagrams, radar images, and correlation plots were generated using Origin2019b. Box plots were generated using Microsoft Excel 2019. Network charts and heat maps were generated using Gephi 0.10 and TBtools-II v.1.120, respectively.

## 3. Results

### 3.1. Soil Characteristics of Four Plots

In accordance with the international system [28], the soil was classified into sand (0.02~2 mm), silt (0.002~0.02 mm), and clay (<0.002 mm). This study showed that the soil of CS1 and CS2 at a depth of 0–30 cm was sandy clay loam, and the soil of CS3 and CS4 at a depth of 0–30 cm was loamy clay based on the soil particle composition (Appendix A, Figure 1). The surface soil permeability of sites 1 and 2 was better than that of sites 3 and 4. The organic matter content of CS1 was higher than that of the other three plots. According to the nutrient classification standard formulated by the second general soil survey of China, the organic matter content of all plots was at the lower level of the national soil classification (grades 4–6). CS1 and CS3 were neutral and slightly acidic soils, and CS4 was neutral soil. The CEC of CS4 was higher than the other plots, and the fertilizer holding capacity was higher. Soil mineral elements at a depth of 30–60 cm were measured (Appendix A). The results showed that the concentrations of total nitrogen, available phosphorus and available iron were significantly higher in CS1 than those in the other three plots. The concentration of available calcium in CS3 was higher than that in the other three plots, and the available magnesium in CS3 and CS4 was higher than that in CS1 and CS2. In general, the organic matter and total nitrogen contents of the four plots were at a low level, the contents of CEC and available potassium were at a medium to low level, and the contents of available phosphorus, available calcium, available magnesium, and available iron were rich.

### 3.2. Physico-Chemical Characteristics of Grapes

The physico-chemical characteristics of the grapes in the 2020 and 2021 vintages were analyzed via one-way and two-way ANOVA to investigate the influence of vintage and plot on characteristics (Table 1). Hundred-grain weight, soluble solid content, titratable acid, and pH showed significant differences between vintages. Hundred-grain weight and pH showed a significant difference between plots. In 2020, the hundred-grain weight of CS2 and CS3 was higher than that of CS1 and CS4. The titratable acid concentration of CS2 was higher than that of the other three plots. The pH of CS1 and CS4 was higher than that of CS2 and CS3. In 2021, the hundred-grain weight of CS4 was higher than that of the other three plots. The pH of CS3 was higher than that of the other three plots. There was no significant difference in soluble solids content between the four plots in any year. The soluble solids content and pH of the four plots was higher in 2020 than that in 2021.

### 3.3. Two-Way ANOVA for Differences in Grape Aroma Compounds between the Plots

A total of 45 free aroma compounds were detected in the 2020 and 2021 grapes. To understand which compounds varied by plot, we performed a two-way ANOVA based on four plots and two vintages. There were 34 compounds that showed a significant difference in concentration between two vintages, 25 compounds that showed a significant difference between the four plots, and 15 compounds that were affected by the combination of vintages and plots (Appendix A).

Regarding the glycosylated aroma compounds, a total of 23 compounds were detected in the 2020 and 2021 grapes. Of these, 19 compounds were found to be significantly different between the two vintages and 8 compounds that were significantly different between the four plots. Three compounds varied based on the combination of vintage and plot (Appendix A).

These plot-influenced compounds were further compared (Figure 2). It was found that out of the nine free-form norisoprenoids and seven monoterpenoids detected in this study, seven and five compounds varied between the plots, respectively. With the exception of *(Z)*-*β*-damascenone and *p*-cymenene, all other free-form norisoprenoids and monoterpenoids showed higher concentrations in the 2021 grapes than in those from 2020. In contrast, of the eight glycosylated aroma compounds that were affected by plot, seven components showed higher concentrations in 2020. 

When CS1 at the top of the slope was compared with CS2 at the bottom, it was observed that many free aroma compounds were higher in CS2 than in CS1 in 2020, of which TDN, *(Z)*-*β*-damascenone, and cadalene showed high levels in CS2 in both vintages. The grapes from CS3 had high concentrations of phenylacetaldehyde, phenylethyl alcohol, hexanal and etc, and the grapes of CS4 had low concentrations of theaspiranes A & B, *p*-cymenene, *p*-cymene and etc. 

Free-form (*E*)-2-hexenal, hexanal, and (*E*)-2-hexen-1-ol were the three compounds with the highest concentration among the free-form aroma compounds in this study (Appendix A), and the latter two had significant differences between the plots, especially (*E*)-2-hexen-1-ol with the greatest variation. The concentration of (*E*)-2-hexen-1-ol was higher in CS1 than in CS2. Similarly, glycosylated *(E)*-2-hexen-1-ol also had a higher concentration in CS1 compared to CS2 and CS3 in 2020. In addition, the concentration of glycosylated *(Z)*-3-hexen-1-ol was lower in CS4 than in the other three plots in both years.

### 3.4. Effect of Soil Characteristics on Aroma Profiles of Grapes

To investigate the influence of soil characteristics on the aroma profiles of the grapes, Pearson’s correlation analysis was performed on soil physico-chemical properties, including 25 free aroma compounds and 8 glycosylated aroma compounds, which were selected via two-way ANOVA. The absolute value of the correlation coefficient between 0.8 and 1.0 was considered a very strong correlation. The absolute value of the correlation coefficient between 0.6 and 0.8 was considered a strong correlation. The absolute value of the correlation coefficient between 0.4 and 0.6 was considered a medium correlation. The result showed that seven free aroma compounds and one bound aroma compound had a strong correlation with soil physico-chemical properties (Figure 3): both free styrene and free *(E)*-3-hexen-1-ol in grapes had a significant positive correlation with soil N, P, and K contents and a negative correlation with pH. Free *(E)*-3-hexen-1-ol was also positively correlated with soil Fe and EC. And free phenylacetaldehyde concentration in grapes was positively influenced by soil Ca and Mg contents. Hexanal in grapes was strongly positively correlated with soil Mg, while *(E)*-2-hexen-1-ol, also a C6 compound, was strongly negatively correlated with Mg. It was found that the bound 3-methyl-1-butanol in grapes was negatively influenced by the soil K and Mg contents.

### 3.5. Physico-Chemical Characteristics of Wines

Alcohol, reducing sugar, total acidity, pH, and volatile acid of wines showed significant difference between vintage and plot (*p* < 0.0001). Wines produced in 2020 were generally higher in alcohol content than those produced in 2021, which was related to higher soluble solids content. In 2020, CS3 wines had higher residual sugar content. There was not much difference between CS2 and CS4 wines. In 2021, the alcohol content of the CS1 and CS3 wines was higher than that of CS2and CS4. In both years, the CS3 wines had the highest total acidity and the lowest pH (Table 2).

### 3.6. Two-Way ANOVA for Differences in Wine Aroma Compounds between the Plots

A total of 74 aroma compounds were detected in the wines after AF (referred to as AF wine) and the wines after MLF (referred to as MLF) in the 2020 and 2021 vintages. To find out which compounds were influenced by plot, a two-way ANOVA was performed for four plots and two vintages (Appendix A). There were 66 compounds in the AF wine and 61 compounds in the MLF wine that showed a statistical difference in concentration between two vintages. In comparison, 58 compounds in the AF wine and 44 compounds in the MLF wine showed a significant difference between the plots, of which 26 and 21 compounds, respectively, had an OAV ≥ 0.1. In addition, 23 compounds in the AF wine and 16 compounds in the MLF wines with an OAV ≥ 0.1 were influenced by the combination of vintage and plot.

These plot-influenced compounds in the wines were further compared (Figure 4). Overall, the concentrations of many aromatic compounds in the AF and MLF wines were higher in 2020 than in 2021, especially the esters that contribute to fruity aroma, such as ethyl acetate, ethyl lactate, ethyl 3-methylbutanoate, and isoamyl acetate. In addition, (*Z*)-*β*-damascenone and linalool were also higher in the AF wines in 2020 than in 2021.

Comparing the plot-influenced compounds in AF and MLF wines, the most significant compounds were 2-heptanol, 1-octen-3-ol, 1-octanol, and isoamyl acetate. Of these, the concentration of isoamyl acetate far exceeded the olfactory threshold (Appendix A), indicating a significant contribution by this compound to the fruity aroma of the wine. Moreover, this compound had an overall high concentration in the CS1 wines in both years and a low concentration in the CS3 wines. However, the concentrations of 2-heptanol, 1-octen-3-ol, and 1-octanol were well below their olfactory threshold (Appendix A) and had no substantial impact on the wine aroma profile, although they showed significant differences between the plots. In the MLF wines, the concentrations of ethyl acetate, ethyl lactate, and ethyl 3-methylbutanoate were all higher than their olfactory threshold (Appendix A), and their difference between the plots appeared to be more pronounced in 2021 than in 2020, which may be related to the higher rainfall in September 2021. 

### 3.7. Effect of Soil Characteristics on Aroma Profiles of Wines

To investigate the impact of soil characteristics on the aroma profiles of wines, Pearson’s correlation analysis was performed on 21 aroma compounds after AF and 10 aroma compounds after MLF in 2020, which were selected by two-way ANOVA (Figure 5). The result showed that 10 and 8 aroma compounds had a strong correlation with soil physico-chemical properties after AF and MLF, respectively. In AF wines, 1-heptanol had a strong–positive correlation with soil Ca and Mg; 2-heptanol and 1-octanol had a strong–negative correlation with N, P, K, Fe, EC, and a strong–positive correlation with soil pH value. 1-octen-3-ol had a strong to extremely strong positive correlation with soil Ca and Mg, respectively. Linalool had a strong positive correlation with Mg, and m-cresol had a strong negative correlation with soil K. 

It is worth noting that ethyl 3-methylbutanoate, ethyl hexanoate, isoamyl acetate, and isopentanoic acid in AF wines had a strong to extremely strong negative correlation with N, P, K, Fe, and EC (except for ethyl 3-methylbutanoate and K) and an extremely strong and strong negative correlation with soil pH (Figure 5), and the four compounds had a significant aroma contributionin the wine aroma due to their concentrations far exceeding the thresholds. In addition, the four aroma compounds showed very significant differences in concentration between the two years; the concentration was higher in 2020 than in the 2021 (Appendix A).

In the MLF wines, the correlation of several compounds such as higher alcohol was consistent with the compounds in the AF wines. Ethyl acetate had a strong to extremely strong negative correlation with N, P, K, Fe, and EC in soil and a strong–positive correlation with soil pH. Phenethyl acetate had a strong to extremely strong positive correlation with N, P, K, Fe, and EC in soil and an extremely strong and strong–negative correlation with soil pH. 

### 3.8. Aroma Profiles of Wines Based on OAV

The aroma compounds in the MLF wines were grouped into 8 categories according to the odor descriptors (Appendix A), including fruity, floral, green, spicy, solvent, fatty, cooked fruit/vegetable, and others. The OAVs of the compounds from the same categories were summed and converted to log10-fold, and radar maps were generated for the two vintages (Figure 6). It was found that the aroma profiles of the wines were characteristic of fruity, floral, and cooked fruit/vegetable odor attributes, and both fruity and floral intensities were generally higher in the 2020 wines than in the 2021 wines. The number of odor descriptors that differed between the plots was higher in 2020 than in 2021. The odor intensities of the fruity, floral, fatty, and cooked fruit/vegetable descriptors differed between plots in both vintages, among which the variation of floral and fruity odor between plots in 2021 was greater than in 2020. In 2020, the CS1 wines had the highest fruity and floral intensities, whereas in 2021, they had the lowest intensities. 

### 3.9. Aroma Profiles of Wines Based on Sensory Analysis

Sensory analysis was carried out on the post-MLF wines from the four plots in 2020 and 2021. Nine dimensions, including floral, red berry, black berry, fruity intensity, toasted/caramel, wood/oak, vanilla/cream, and smoky/spicy were scored and averaged to produce a radar map (Figure 7).

In general, there was little difference in the sensory characteristics of the post-MLF wines between the two years in the categories of medium fruity intensity, red fruity, and blackberry flavor. In the 2020 wines, the sensory characteristics of floral, black berry, toasted/caramel and vanilla/cream flavor were more pronounced and the intensity of green and red berry were lower compared to the 2021 wines. In 2020, the sensory characteristics varied between the plots, and the wine from CS3 showed a better aroma quality. In 2021, the sensory characteristics did not show any significant difference between the plots.

### 3.10. Alpha Diversity Analysis of Aroma Compounds

To quantify the aroma diversity of grapes and wines, the Shannon–Wiener index was used to evaluate grape aroma compounds and wine aroma compounds in two vintages (Figure 8). The diversity of CS1 and CS4 grapes was higher than CS2 and CS3 grapes in 2020, whereas the diversity of CS1 and CS2 grapes was higher than CS3 and CS4 grapes in 2021. The diversity of grape aroma was lower in 2020 than in 2021 in the same plots, except for CS4. For wines, the diversity of those from CS2 was lower than that of those from the other three plots in 2020, while the diversity of CS1 was the highest and CS4 was the lowest of the four plots in 2021. The wine aroma diversity of 2020 was lower than that of 2021 in CS1 and CS2, while the wine aroma diversity of CS3 and CS4 did not show any difference between the two vintages. Overall, the CS1 wines had a rich aroma diversity in both vintages, whereas the CS2 wines had low Shannon–Wiener index. From yearly perspective, the difference in the Shannon–Wiener index was more pronounced in 2021 than in 2020. 

## 4. Discussion

The four Cabernet Sauvignon plots studied are located on a gentle slope with a gradient of 0.3%, where the influence of rainfall is relatively greater than that of sunlight and temperature. Practical experience shows that there are certain differences in the aromatic characteristics of the wines from these four plots. This study investigated the soil characteristics and aroma compounds in grapes and wines from these plots. The results clarified the main aroma components that differed between the plots and dissected their correlation with soil physicochemical properties. Based on the differences in rainfall between the two years, it is assumed that high rainfall will make the differences in aroma profiles between the plots more apparent. 

### 4.1. Variation of Aroma Compounds in Grapes and Wines between the Plots

Water deficit in the special zone helps to increase the content of C6 alcohols in grapes [23]. Researchers have found that reducing water supply improved the content of 1-hexanol in grapes, which was associated with the up-expression of two genes, *Vv*LOX and *Vv*HPL, in the biosynthetic pathway of hexanol [29]. The accumulation of (*E*)-*β*-damascenone and *β*-ionone in grapes was strongly influenced by temperature, humidity, sunshine duration, frost-free days, etc. [30]. At the same sugar concentrations, higher temperatures contributed to lower monoterpene levels in white aromatic grape varietals, resulting in reduced aromatic intensity [31]. In this study, the grapes had higher concentrations of free-form C6 compounds and lower norisoprenoids concentrations of monoterpenoids in 2020 than in 2021, which may be due to the lower rainfall and more sunshine in 2020. Interestingly, glycosylated aroma compounds had higher concentrations in 2020, which is different from the concentrations of free-form aroma compounds.

The grape aroma components affected by the plot were selected via two-way ANOVA (Figure 2). Among these components, monoterpenoids and norisoprenoids were the most affected, with most of them exhibiting plot differences. Norisoprenoids and monoterpenoids are synthesized via carotenoid metabolism and the 2-C-methyl-D-erythritol-4-phosphate (MEP) pathway [10,32]. Norisoprenoids and monoterpenoids often contribute to the floral and fruity odor of grapes and wines, especially in non-aromatic grape varieties such as Cabernet Sauvignon. The concentration of norisoprenoids in grapes is regulated by sun exposure [11]. In addition, Yuan et al. observed that low nitrogen status was associated with low β-damascenone content in wines [12]. In our study, most norisoprenoid and terpene compounds, especially theaspirane B, were at lower concentrations in wines from CS4 than in those from the other three plots in both vintages, possibly because of the lower soil N content in CS4. Four green leaf odor components, hexanal, (*E*)-3-hexen-1-ol, (*Z*)-3-hexen-1-ol and (*E*)-2-hexen-1-ol, showed differences between the plots in this study, especially (*E*)-2-hexen-1-ol, which had the greatest variation. Water deficit favors the increase of the content of C6 alcohols in grapes [23]. The concentration of (*E*)-2-hexen-1-ol was higher in CS1 than in CS2. Similarly, the concentration of glycosylated *(E)*-2-hexhen-1-ol in CS1 was higher compared to CS2 and CS3 in 2020, which may be related to the fact that the CS1 plot was located at the top of the slope. The location of CS1 facilitates water permeation.

Glycosically bound aroma compounds are composed of components with free hydroxyl group(s), mainly aliphatic alcohol derivatives (higher alcohols, C6 compounds), terpenoids, norisoprenoids, and benzenoids in grapes [33]. Researchers have found that sunshine is beneficial for the accumulation of glycosylated terpenes [34]. In Agiorgitiko vines, limited water supply can increase the levels of the glycoconjugates of the main aroma compounds [35]. The previous reports explained why most of the bound aroma compounds had a higher concentration in 2020, when there was more abundant sunshine and less rainfall.

The aroma components affected by the plot were selected from AF and MLF wines, respectively, via two-way analysis of variance and consisted mainly of higher alcohols and fatty acid ethyl esters (Figure 4), which was different from previous studies. Slaghenaufia et al. studied the differences in wine aroma of different plots in a vineyard in the Valpolicella wine region and found that the compounds causing the differences between plots were mainly benzenoid compounds, terpenes, and norisoprenoids [36]. In our study, the compounds of most concern are ethyl acetate, ethyl lactate, ethyl 3-methylbutanoate, and isoamyl acetate. Esters are synthesized mainly by yeast but can also be synthesized by lactic acid bacteria. They play a central role in characterization of fruity aromas, with ethyl esters contributing more [37]. It is believed that in non-aromatic grape varieties such as Cabernet Sauvignon, fruity odors are generated mainly by ethyl esters such as ethyl isobutyrate, ethyl butyrate, ethyl 3-methylbutyrate, ethyl hexanoate, ethyl octanoate, etc. [33]. In the MLF wines in 2020, the concentration of higher alcohols such as 1-heptanol, 2-heptanol, 1-octen-3-ol, and 1-octanol in CS1 was at a lower level and the concentration of ethyl 3-methylbutanoate and isoamyl acetate was at a higher level, contributing to more intense floral and fruity odor in CS1 (Figure 6). In MLF wines in 2021, the concentration of 1-octen-3-ol, ethyl acetate and ethyl lactate were at a higher level in CS1 and the concentration of ethyl 3-methylbutanoate and isoamyl acetate was at a higher level in CS4, leading to the higher aroma quality in CS4.

Overall, isoamyl acetate, which has a very high OAV in the wines in this study, was present at a high concentration in the CS1 wines in both years and a low concentration in the CS3 wines. This compound contributes to the banana-like fruity note of the wine. According to the radar map of the added OAV of the aroma categories, the intensity of fruity and floral of wines in 2020 was higher than that in 2021, and the intensity of solvent was lower in 2020. It was found that rainfall can affect the maturity of grapes and excessive rainfall can reduce the sugar content of grapes, increase acidity, and dilute the flavor of grapes [38]. From the content of soluble solids and titratable acid in the grapes from the four plots, the maturity of grapes in 2020 was better than that in 2021 except acidity in CS2, and the CS2 grapes had higher acidity in 2020 and lower acidity in 2021 (Table 1). The higher maturity and aroma quality of the wines in 2020 may be related to its lower rainfall compared to 2021. Studies have shown that as the maturity of Cabernet Sauvignon increases, green (vegetable) flavor decreases and red berry aroma shifts to blackberry aroma in the resulting wines [39]. In addition, in the MLF wines, ethyl esters of acetate, lactate, and 3-methylbutanoate with high OAVs showed a greater difference in concentration between the plots in 2021 than in 2020, suggesting that high rainfall would accentuate the differences in vineyard terrain. 

Sensory analysis revealed a significant difference between the two vintages and indicated that the fruity and floral aroma characteristics were better in 2020, which was consistent with the aroma profile based on OAV and may be due to the lower rainfall in 2020. However, there was a contradictory result, namely that the sensory analysis showed greater difference between plots in 2020, while the aroma profile based on OAV showed a greater difference between plots in 2021. Furthermore, the aroma quality of CS3 wines was higher than other plots based on sensory analysis, while CS1 wine was the best based on OAV. It is commonly known that wine aroma is not only the simple complex of individual aroma compounds, but also related to the interaction between aroma compounds and the influence of the wine matrix, such as polyphenols, proteins, carbohydrates, alcohols, etc. Researchers have found that glucose in wine can increase the release of volatile compounds [40,41,42,43,44]. The higher reducing sugars and other nonvolatile components in CS3 wines in 2020 may have influenced the sensory aroma profile.

### 4.2. Correlation between Some Aroma Compounds in Grapes and Wines and Soil Physico-Chemical Properties

The physical loss of soil through mechanical cultivation and displacement through erosion is likely to be exacerbated by heavy rainfall [45]. Although the four plots in this study are located on the same slope, due to the temperate monsoon climate and concentrated rainfall in the production area, there are certain differences in soil physico-chemical properties between the plots. It is known that the physico-chemical properties of the soil influence the growth of roots of the vine and thus the quality of the grapes. Among all characteristics, nitrogen is the most important nutrient element that restricts plant growth. The nitrogen content of the soil affects the concentration of nitrogenous compounds in the grapes, such as total nitrogen, amino acids, ammonium salts, and assimilable nitrogen [46]. Nitrogen influences the vine vigor, yield, and berry size and has an effect on the major metabolites (sugars, organic acids) and secondary metabolites (phenolic compounds, flavors and aroma precursors) of grapes [47]. In addition, phosphorus is one of the most important elements for plant growth and reproduction. Phosphorus plays an important role in improving the uptake and transformation of nitrogen and can affect flower bud differentiation and fruit development as well as improving the uptake capacity of the root system in the plant [48,49]. Most of the potassium elements that can be absorbed and utilized by grapes come from the soil, and the potassium content in plants is high and similar to that of nitrogen. During the growth period of wine grapes, there is a high demand for potassium, which can enhance photosynthesis, improve nitrogen metabolism and carbohydrate metabolism, improve the rate of water use absorbed by the grapes, and increase grape stress resistance and disease resistance [50]. In this study, there was a positive correlation between free theaspirane B concentration in the grapes and soil K and N contents, suggesting that the lower theaspirane B concentration in the CS4 grapes in both vintages may be associated with the lower soil K and N contents in CS4. 

Available iron is one of the elements that make up chlorophyll, which is involved in photosynthesis and respiration, and iron deficiency usually results in yellowing of new shoots and young leaves [51,52]. Electrical conductivity value (EC) is a parameter of water-soluble salts in the soil, which is a factor that determines whether salt ions in the soil will limit plant growth; too high or too low a concentration can hinder plant growth [53]. This study showed that free *€*-3-hexen-1-ol, a component of green leaf odor, had a strong positive correlation with soil N, P, K, Fe, and EC and a strong negative correlation with soil pH. CS1 soil had higher levels of N, P, K, and Fe, corresponding to a higher concentration of (*E*)-3-hexen-1-ol in the CS1 grapes.

Previous studies have demonstrated the effects of soil physico-chemical properties on wine aroma quality. Nitrogen is the most abundant soil-derived macronutrient in a grapevine and plays an important role in fermentative microorganisms [46]. The present study showed that higher alcohol concentration was negatively correlated with soil N content, which may explain why wines from CS1, with its higher soil N content, contained a lower concentration of higher alcohols. The relationship was also consistent with the research showing that low YAN results in high content of higher alcohols [46].

Compounds that promoted wine aroma quality, such as ethyl 3-methylbutanoate, ethyl hexanoate, isoamyl acetate, and phenethyl acetate, were positively correlated with soil N, P, K, Fe, and EC, but negatively correlated with soil pH. Conversely, compounds with certain negative effects on wine flavor, such as 2-heptanol and 1-octanol, were strongly negatively correlated with soil N, P, K, Fe, and EC and strongly positively correlated with soil pH. The result was consistent with a previous report that N fertilization of a Riesling vineyard increased 1-butanol, trans-3-hexen-1-ol, benzyl alcohol, and most of the esters in wines [54].The content of N, P, and Fe of CS1 was significantly higher than that of the other three plots, which may be the reason why the wine aroma quality of CS1 was better than that of the other three plots. Soil pH may be important in defining the unique ageing characteristics of a particular vineyard [12]. Given the present results, it is suggested that the range of soil pH was from 6.46 to 7.05, which seemed to produce a pleasant odor with lower soil pH in a certain range.

As previously reported, the water status of the grapevine is an important determinant driver of terroir expression. It depends on climatic conditions (rainfall and reference evapotranspiration) and soil type (soil water holding capacity, SWHC). Wine aromatic typicity is strongly influenced by vine water status [55]. Based on the above research, it has been concluded that soil physico-chemical properties play an important role in wine aroma, and different vintages may also affect soil physico-chemical properties and thus wine aroma quality. In 2020, when there was less rainfall than in 2021, the physico-chemical properties of CS1 resulted in a better wine aroma profile based on OAVs. In 2021, when there was a great deal of rain, the soils of CS3 and CS4 with loamy clay texture had higher viscosity and better fertility protection than those of CS1 and CS2 with their sandy soil layers, and the wine produced from CS4 had better aroma quality with higher soil Fe content than CS3. On the other hand, compared to CS2 at the bottom, CS1 at the top of the slope had better soil, so the fertility protection was poor and the wine aroma quality was poor. It was speculated CS1 and CS2 were more likely affected by rainfall.

### 4.3. Quantifying the Complexity of Aroma Compounds

In systems biology, α-diversity refers to the assessment of the diversity of a single ecosystem or sample [26]. In this study, in order to quantify the complexity of aroma compounds in grapes and wines from different plots, we introduced the α-diversity index commonly used in systems biology research. Here, we use the Shannon–Wiener index to comprehensively evaluate the quantity and concentration of aroma compounds detected in grapes or wines from each plot [27]. Overall, the aroma compound diversity of CS1 and CS2 wines was affected by vintage, while the aroma compound diversity of CS3 and CS4 wines was not found to differ between vintages. Aroma diversity varied between plots from one vintage to another. The amplitude of variation of the Shannon–Wiener index was more pronounced in 2021 than in 2020. As mentioned above, high rainfall in 2021 could have led to a more pronounced vineyard terrain effect. Soil microbial diversity may also lead to more efficient mineralization of soil nutrients [56]. In the vineyards of southern Australia, the soil fungal community plays an important role in wine aroma [57]. It has therefore been speculated that aroma diversity may be related to soil microbial diversity. Liu et al. [58] proved that the functional diversity of microorganisms at the Chihuahuan Desert Ranch in northern New Mexico, USA was lower in the summer drought test site than in the summer and spring controls. In this context, the lower rainfall in 2020 may reduce microbial diversity and thus affect the aroma diversity. As previously reported [59], microbial diversity was higher in soils with higher organic matter content. In this study, the aroma compound diversity of CS1 wines was the highest among the four plots in 2020 and 2021, which may be related to the higher microbial diversity caused by higher organic matter in CS1. 

## 5. Conclusions

In this study, we investigated grape and wine aroma compounds and the terrain of four plots on the same slope so as to find the deep relationship between the two and thus provide a new way to improve wine aroma quality in this region. One-way ANOVA, two-way ANOVA, and Pearson’s correlation analysis were used to find the elements that influenced the aroma profiles. In summary, the variation in aroma compounds was greater between vintages than between plots. Based on two-way ANOVA, the most plot-variant aroma compounds were identified, including 25 free and 8 bound components in grapes and 21 and 10 components in AF and MLF wines, respectively. Of these, most of the free norisoprenoids, monoterpenoids, and C6 compounds in grapes varied between plots. The concentration of free and bound (*E*)-2-hexen-1-ol was higher in CS1 than in CS2, which may be related to the location of CS1 at the top of the slope with better water permeability. Higher alcohols and esters in AF and MLF wines were most affected by the plots, including 1-heptanol, 2-heptanol, 1-octen-3-ol, ethyl 3-methylbutanoate, isoamyl acetate, etc. Wines from the four plots showed a significant difference in floral and fruity aroma attributes, which were mainly related to ethyl esters with high odor activity values. In addition, the variation of these compounds between the plots was more pronounced in the 2021 vintage with more rainfall compared to the 2020 vintage. Pearson’s correlation analysis showed that concentrations of (*E*)-3-hexen-1-ol in grapes and ethyl 3-methylbutanoate, ethyl hexanoate, isoamyl acetate, isopentanoic acid, and phenethyl acetate in wines were strongly positively correlated with the concentrations of N, P, K, Fe, and EC in the soil, but negatively correlated with soil pH. And the result showed the esters contributing to floral and fruity odor were positively correlated with N, P, K, Fe, and EC in soil and negatively correlated with soil pH, while the situation was exactly reversed for higher alcohols contributing to off-odor in wines. Interestingly, the MLF wines from CS1, with its higher soil N, P, and Fe contents, showed higher aroma quality in 2020 but lower aroma quality in 2021, which may be related to the negative effect of fertility loss caused by more rainfall in 2021. The α-diversity of aroma compounds was also calculated, and it was speculated that the variations in α-diversity between vintages and between plots were caused by the differences in rainfall and soil organic matter. This study was the first to dissect the influence of vineyard terrain on the aroma profile of grapes and wines in the Jieshi Mountain region with a temperate monsoon climate and provides some guidance for improving vineyard management and the quality of grapes and wines. The research can be complemented in the future by expanding the scale of wine making, increasing the number of grape varieties, and monitoring the changes in grapevine ageing.

## Figures and Tables

**Figure 1 foods-12-02668-f001:**
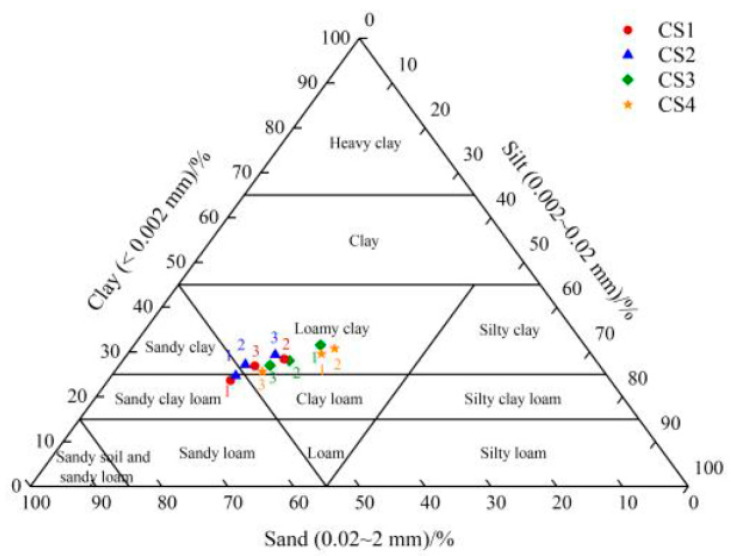
The soil particle composition of four plots: 1, 2, and 3 represent depths of 0–30 cm, 30–60 cm, and 60–90 cm, respectively.

**Figure 2 foods-12-02668-f002:**
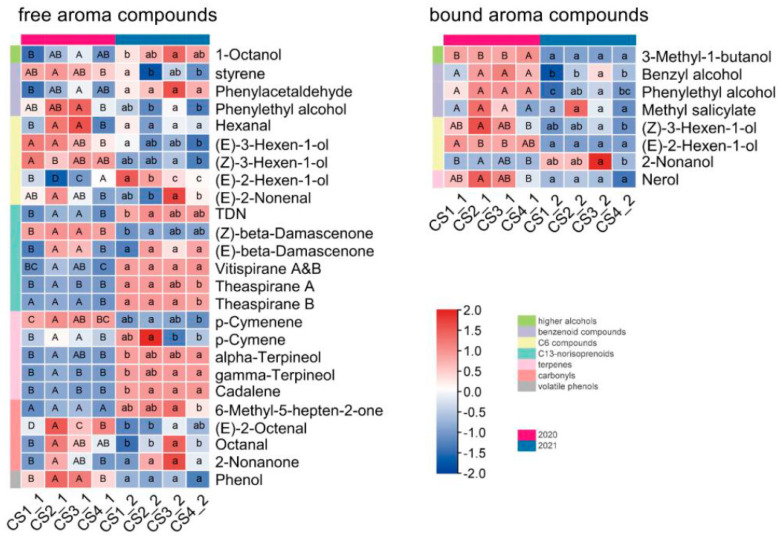
Heat maps for statistical difference in the concentration of aroma compounds between the plots in grapes. The components listed in this figure were shown via two-way ANOVA to be significantly influenced by plot across the two vintages (Appendix A). The concentrations were converted to log2 fold for standardization. The capital letters represent the differences between the four plots in 2020 determined via one-way ANOVA (*p* ≤ 0.05), and the lowercase letters represent the differences in 2021.

**Figure 3 foods-12-02668-f003:**
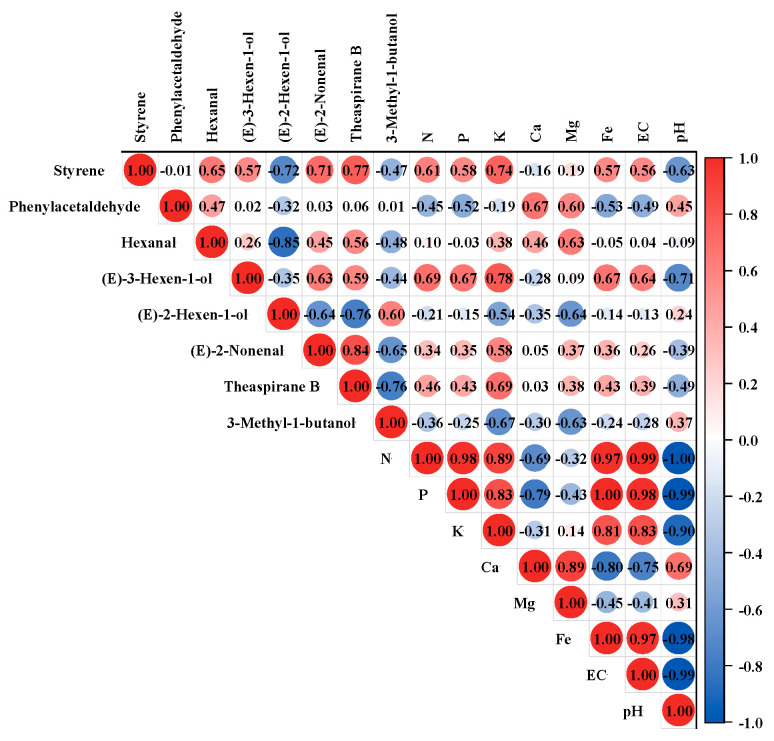
Pearson’s correlation coefficients between the concentration of aroma compounds in grapes and the physico-chemical properties of soil. The data from soils at a depth of 30–60 cm was used for analysis. 3-Methyl-1-butanol was the only bound aroma compound.

**Figure 4 foods-12-02668-f004:**
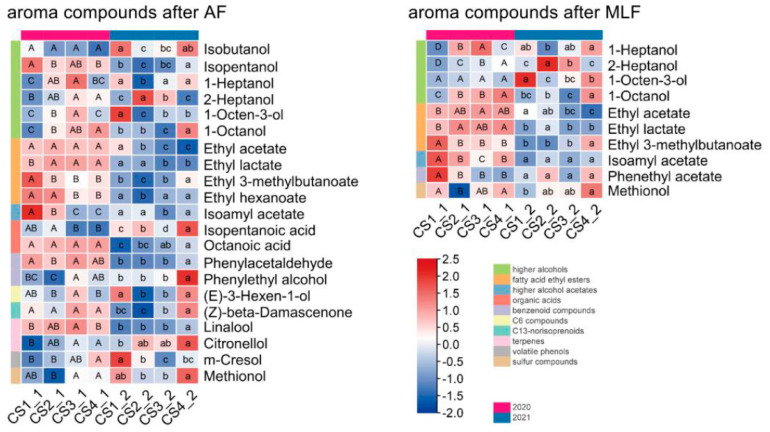
Heat maps for statistical difference in the concentration of aroma compounds between plots in wines. AF: Alcohol fermentation; MLF: Malolactic fermentation. The components listed in this figure were shown via two-way ANOVA to be significantly influenced by plot across two vintages (Appendix A). The concentrations were converted to log2 fold for standardization. The capital letters represented the differences between four plots in 2020 by one-way ANOVA (*p* ≤ 0.05) and the lowercase letters represented the differences in 2021.

**Figure 5 foods-12-02668-f005:**
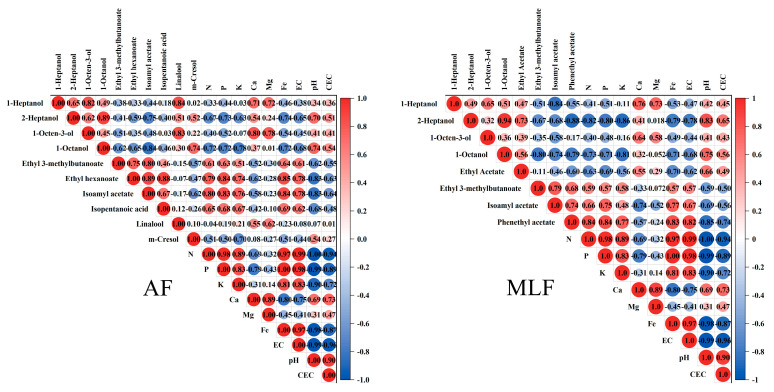
Pearson’s correlation coefficients among the concentration of aroma compounds of wines and the physico-chemical properties of soil. Data from soils at a depth of 30–60 cm was used for analysis.

**Figure 6 foods-12-02668-f006:**
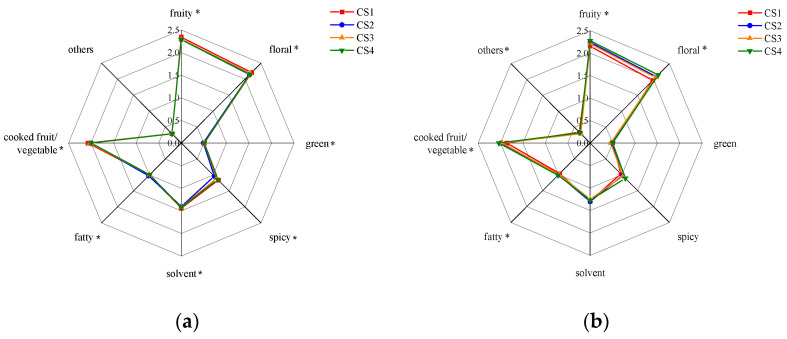
Aromatic categories calculated by adding the odor activity values of the compounds grouped in each one. (**a**) 2020; (**b**) 2021. The values were converted to log10 fold for visualization. * indicates that the values of one-way ANOVA (*p* ≤ 0.05) varied significantly between the four plots.

**Figure 7 foods-12-02668-f007:**
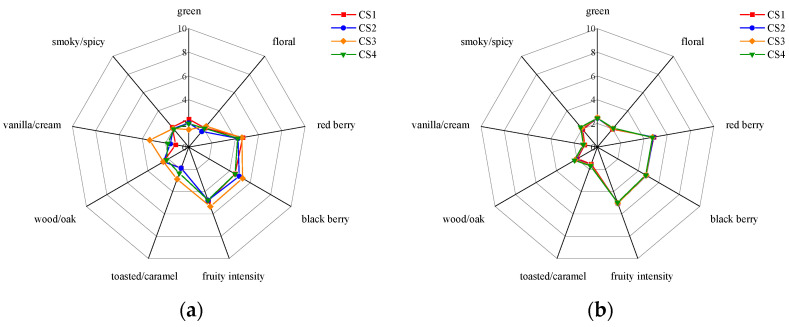
Sensory analysis results of wines after MLF in the year of 2020 and 2021. (**a**) 2020; (**b**) 2021.

**Figure 8 foods-12-02668-f008:**
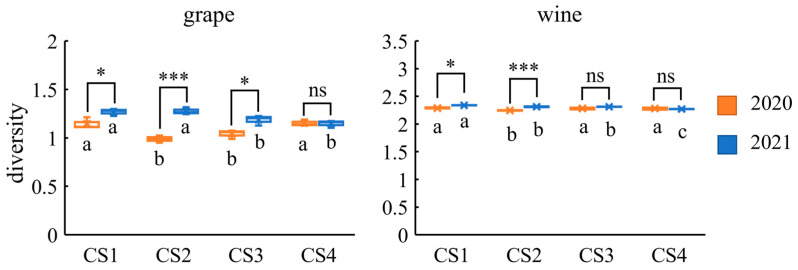
Variation of Shannon–Wiener index of grape and wine aroma compounds in two vintages. Different letters indicate significant differences between plots in the same year (*p* < 0.05). * and *** indicate significant differences between vintages in the same plot at *p* < 0.05 and 0.001 based on one-way ANOVA, respectively. ns indicates no significance.

**Table 1 foods-12-02668-t001:** Physico-chemical characteristics of grapes from four plots over two vintages.

Characteristics	Vintage	CS1	CS2	CS3	CS4	V	P	V × P
hundred-grain weight (g)	2020	146.8 ± 0.7 b	163.7 ± 4.2 a	166.0 ± 5.5 a	151.6 ± 5.3 b	*	****	****
	2021	147.7 ± 3.5 c	126.3 ± 9.5 d	161.4 ± 1.6 b	172.6 ± 2.8 a			
soluble solids content (Brix)	2020	24.0 ± 0.9 a	23.3 ± 0.1 a	23.4 ± 0.8 a	23.6 ± 0.7 a	***	ns	ns
	2021	22.2 ± 1.8 a	21.2 ± 1.1 a	22.4 ± 0.8 a	21.7 ± 1.0 a			
titratable acid (g tartaric acid/L)	2020	3.7 ± 0.4 b	5.4 ± 1.1 a	3.9 ± 0.1 b	4.1 ± 0.3 b	***	ns	**
	2021	5.0 ± 0.4 ab	4.7 ± 0.2 b	5.2 ± 0.3 a	5.5 ± 0.1 a			
pH	2020	3.87 ± 0.01 a	3.65 ± 0.01 c	3.77 ± 0.02 b	3.84 ± 0.03 a	****	****	****
	2021	3.41 ± 0.01 d	3.43 ± 0.01 c	3.62 ± 0.01 a	3.58 ± 0 b			

Note: Different letters represent significant differences between plots in the same vintage based on one-way ANOVA (*p* ≤ 0.05). V: vintages; P: plots; V × P: vintages and plots. *, **, ***, and **** indicate significance at *p* < 0.05, 0.01, 0.001 and 0.0001 based on two-way ANOVA, respectively. ns indicates no significance.

**Table 2 foods-12-02668-t002:** Physico-chemical properties of wines at the end of malolactic fermentation.

Characteristics	Vintage	CS1	CS2	CS3	CS4	V	P	V × P
alcohol (%)	2020	13.1 ± 0.02 a	12.7 ± 0.01 c	12.6 ± 0.02 d	12.9 ± 0.0 b	****	****	****
	2021	12.1 ± 0.0 b	11.6 ± 0.02 d	12.1 ± 0.02 a	11.8 ± 0.0 c			
reducing sugar (g/L)	2020	2.9 ± 0.05 c	3.0 ± 0.05 bc	6.5 ± 0.14 a	3.1 ± 0.05 b	****	****	****
	2021	3.5 ± 0.05 a	3.2 ± 0.09 b	3.4 ± 0.08 a	3.2 ± 0.0 b			
total acidity (g tartaric acid/L)	2020	5.9 ± 0.0 c	6.0 ± 0.0 b	7.2 ± 0.05 a	5.7 ± 0.05 d	****	****	****
	2021	5.9 ± 0.0 c	6.0 ± 0.05 b	6.2 ± 0.0 a	5.9 ± 0.0 c			
pH	2020	3.71 ± 0.01 b	3.63 ± 0 c	3.42 ± 0.01 d	3.75 ± 0 a	****	****	****
	2021	3.77 ± 0 a	3.65 ± 0 b	3.59 ± 0.01 c	3.65 ± 0 b			
volatile acid (g acetic acid/L)	2020	0.5 ± 0 a	0.5 ± 0.01 a	0.6 ± 0 a	0.5 ± 0 b	****	****	****
	2021	0.5 ± 0 a	0.5 ± 0.01 b	0.4 ± 0 c	0.4 ± 0 c			

Note: Different letters represent significant differences determined via one-way ANOVA (*p* ≤ 0.05). V: vintages; P: plots; V × P: vintages and plots. **** indicates significance at *p* < 0.0001 determined via two-way ANOVA. ns indicates no significance.

## Data Availability

Data is contained within the article or Appendix A.

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
