# Peer review of "Differences in Aroma Profile of Cabernet Sauvignon Grapes and Wines from Four Plots in Jieshi Mountain Region of Eastern China"

_foods, 2023, doi:10.3390/foods12142668_

Round 1
Author Response
Dear reviewer,
Thanks for your review. The cover letter was enclosed.
Best regards,
Zhuo Chen

Reviewer 2 Report
The present paper explores the relationship between the ‘Cabernet Sauvignon’ wine style and terrain. The paper is interesting and well written I only have some minor remarks.
Par. 2.3.4 please reduce this paragraph, unless the method was firstly described in this study.
Conclusion should be revised since this section simply reports the obtained data
Tables 1 and 2. please perform a t-test
Did the authors perform a sensory analysis with panelists?
The authors calculated the alpha diversity analysis of aroma compounds, this result should be better discussed also referring to the possible role of microbes and soil characteristics in its definition.
Author Response

(The authors gave the same response as above.)

Reviewer 3 Report
Abstract
Ethyl esters instead of eaters
Citation 1 refers mostly to the composition of the soil but it does not link to the composition in wines. Add also a reference that measures both.
Line 48. Add reference about wine quality
Material and Methods
How many days were the fermentations?
Results
Table 1. Why one-way Anova? Better to perform with both variables, plot and vintage, especially if you are doing comparisons between all for the correlations.
Figure 4, since different letters only apply to each vintage and not to both vintages, it would make it easier to have them with a separation.
Also, are the *, ** etc, for one single vintage or are significant differences for both. If it is for one vintage, specified, and provide the level for 2020 and 2021. If they are for both vintages, why not to have the significance in letters for both.
Figure 5. Pearson’s correlation coefficient includes all data. Whereas that is a good approach to understand on the overall between different variables. Did the authors think of performing correlations for each plot separately and evaluate the differences?
Nonetheless, a Multiple Factor Analysis (MFA) is more adequate analytical strategy to analyse the data as it is a multiblock dataset where you have different plots, different vintages, different sets of measurements.
Author Response

(The authors gave the same response as above.)
